# Mechanistic Insights into Nitrite Degradation by Metabolites of *L. plantarum* A50: An LC-MS-Based Untargeted Metabolomics Analysis

Jiangbo An [1], Lin Sun [2], Mingjian Liu [1], Rui Dai [1], Qiang Si [1], Gentu Ge [1], Zhijun Wang [1] and Yushan Jia [1,*]

[1] Key Laboratory of Forage Cultivation, Processing and High Efficient Utilization, Ministry of Agriculture, China, Key Laboratory of Grassland Resources, Ministry of Education, China, College of Grassland, Resources and Environment, Inner Mongolia Agricultural University, Hohhot 010019, China; an474608345@sina.com (J.A.); liumj_nm@163.com (M.L.); 18747659997@163.com (R.D.); siqiang_nm@126.com (Q.S.); gegentu@163.com (G.G.); zhijunwang321@126.com (Z.W.)

[2] Inner Mongolia Academy of Agricultural & Animal Husbandry Sciences, Hohhot 010031, China; sunlin2013@126.com

* Correspondence: jys_nm@sina.com

**Abstract:** Nitrites are universally acknowledged natural toxic substances that frequently lead to poisoning in humans and animals. During fermentation, certain microorganisms utilize a portion of the nitrogen element and reduce nitrates to nitrites through specific metabolic pathways. In this study, a highly effective lactic acid bacterial strain, *Lactiplantibacillus plantarum* A50, was isolated and screened from alfalfa silage for its remarkable ability to degrade nitrites. *L. plantarum* A50 exhibits exceptional nitrite removal capacity, with a degradation rate of 99.06% within 24 h. Furthermore, *L. plantarum* A50 demonstrates normal growth under pH values ranging from 4 to 9 and salt concentrations of 5%, displaying excellent tolerance to acidity, alkalinity, and salinity. Additionally, it undergoes fermentation using various carbon sources. Within the first 6–12 h of culture, *L. plantarum* A50 primarily achieves nitrite degradation through non-acidic processes, resulting in a degradation rate of 82.67% by the 12th hour. Moreover, the metabolites produced by *L. plantarum* A50 exhibit a synergistic interaction with acidity, leading to a nitrite degradation rate of 98.48% within 24 h. Notably, both *L. plantarum* A50 and MRS broth were found to degrade nitrites. Consequently, a non-targeted metabolomic analysis using LC-MS was conducted to identify 342 significantly different metabolites between *L. plantarum* A50 and MRS broth. Among these, lipids and lipid-like molecules, organic acids and derivatives, organic oxygen compounds, and organoheterocyclic compounds emerged as the main constituents. Lipids and lipid-like molecules, derivatives of glucose and galactose, amino acids and their derivatives, as well as organoheterocyclic compounds, are likely to play a role in nitrite elimination. Through the enrichment analysis of differential metabolic pathways using KEGG, nine distinct pathways were identified. These pathways provide essential nutrients, maintain cellular structure and function, participate in substance transport, regulate metabolic activities, and enhance resistance against pathogenic microorganisms in *L. plantarum* A50.

**Keywords:** *Lactiplantibacillus plantarum*; physiological characteristics; metabolic products; nitrites

## 1. Introduction

Nitrites are a commonly employed food additive in the food industry, serving primarily as a preservative and flavor enhancer [1]. Possessing antimicrobial and antioxidative properties, they effectively extend the shelf life of food products and enhance their taste [2,3]. Nitrites inhibit bacterial growth and can react with myoglobin in meat to form stable nitrite–myoglobin complexes, resulting in the vibrant red color of meat [4]. Additionally, research has shown that nitrites and nitrates found in certain foods and diets can be metabolized into nitric oxide, contributing to cardiovascular health and cellular protection [5,6]. Such an

application of nitrites is particularly widespread in the processing of meat products, vegetables, and animal feed, including the fermentation of sausages, pickles, and silages [3,7]. Despite these advantages, nitrites come with certain risks and potential hazards. Experimental evidence has indicated that under specific conditions, nitrites can transform into carcinogenic nitrosamines, posing potential threats to human health [8,9]. Moreover, excessive consumption of nitrites can be detrimental to human health, leading to conditions such as hypertension and glioma [10,11]. Given these considerations, a crucial question arises as to whether existing methods can effectively reduce nitrite levels while guaranteeing fermentation quality.

Lactic acid bacteria commonly prevail as the primary microorganisms in the fermentation procedure [12]. Lactic acid bacteria, as a probiotic, possess immense potential in preventing human health-related ailments and aiding in food preservation and fermentation [13]. Throughout the fermentation process, lactic acid bacteria effectively hinder the growth of other microorganisms by reducing the pH levels through the production of acid, thus effectively preventing food spoilage. Moreover, lactic acid bacteria are capable of generating beneficial compounds, such as flavor-enhancing substances, thereby enhancing the taste and overall quality of the food [14]. Furthermore, numerous studies have demonstrated the ability of lactic acid bacteria to eliminate nitrite. Kim et al. [15] scrutinized the effects of lactic acid bacteria on kimchi and discovered that they can directly degrade N-nitrosodimethylamine, subsequently resulting in the reduction of nitrite levels. Lactic acid bacteria achieve this by reducing nitrite either through acid production or nitrite reductase in fermentation products [16,17]. By glycolysis, lactic acid bacteria produce organic acids, including lactic acid and acetic acid. These organic acids possess certain reducing properties, while nitrite exhibits strong oxidizing properties. Consequently, a redox reaction occurs between nitrite and organic acids during fermentation, effectively reducing the nitrite content. Additionally, Paik et al. [18] discovered that *Lactiplantibacillus plantarum*, *Levilactobacillus brevis*, *Latilactobacillus curvatus*, and *Lactobacillus serans* possess the capacity to synthesize nitrite reductases. Depending on the reactants and coenzymes, these nitrite reductases can be classified as CuNiRs, cd1NiRs, ccNiRs, and FdNiRs [19]. Aside from acid and enzyme degradation pathways, other compounds such as flavonoids, polyphenols, and ascorbic acid have also demonstrated their influence on nitrite scavenging. Guo et al. [20] discovered a notable association between flavonoids and their capacity to act as antioxidants and eliminate nitrites. Ascorbic acid functions as a reducing agent that converts nitrite to NO, while undergoing oxidation to dehydroascorbic acid. It is important to note that the degradation of nitrite is minimal at pH levels above 4.0 [21,22]. Moreover, the addition of ascorbic acid during fermentation not only reduces nitrite content but also effectively inhibits the formation of nitrosamines [23]. In another study, Seo et al. [24] investigated the production of extracellular polysaccharides by *Lactobacillus plantarum* YML009. They observed a nitrite-scavenging rate of 43.93%, demonstrating the greater ability of extracellular polysaccharides to degrade nitrite. Furthermore, the higher sugar content in these polysaccharides corresponds to a higher rate of nitrite radical scavenging. Therefore, lactic acid bacteria exhibit diverse mechanisms for nitrite degradation, urging further exploration of their potential in this regard through detailed and comprehensive means.

In recent times, metabolomics has witnessed a growing application in elucidating the metabolic mechanisms of lactic acid bacteria [25,26]. The high resolution and sensitivity associated with LC-MS permit accurate identification of lactic acid bacteria metabolites. In comparison to other analytical techniques, LC-MS provides more intricate details about metabolites and can differentiate between similar compounds, thus assisting in the determination of lactic acid bacteria metabolite structures and species [27]. LC-MS-based untargeted metabolomics has the potential to identify new compounds and metabolic pathways in lactic acid bacteria metabolites. A comprehensive analysis of metabolomics can result in the discovery of new bioactives, including active molecules, antimicrobial substances, and antioxidants. The outcomes of this discovery can have beneficial effects

in the food, nutraceutical, and pharmaceutical industries [28]. Therefore, LC-MS-based untargeted metabolomics appears to be a dependable method for metabolic profiling.

In the current research, we conducted a meticulous screening of alfalfa silage, where a remarkable lactic acid bacterial strain (*Lactiplantibacillus plantarum* A50) exhibiting exceptional capacity for nitrite degradation was discovered. The efficiency of nitrite degradation by *L. plantarum* A50 was an astounding 99.06% within a mere 24 h incubation period. To delve deeper into the metabolic intricacies and furnish more precise, comprehensive, and detailed information, we propose employing LC-MS non-targeted metabolomics analysis to unravel the metabolites produced by *L. plantarum* A50. This analysis will enlighten us on the biological properties, nitrite-degrading functionality, and potential applications of this strain of lactic acid bacterium. By cultivating *L. plantarum* A50 in MRS broth, we have successfully identified a plethora of differentially accumulated metabolites and have meticulously scrutinized their implications on biological pathways and nitrite degradation. The findings of this study present additional opportunities for employing *L. plantarum* A50 in regulating nitrite levels in fermented products.

## 2. Materials and Methods

### 2.1. Preservation and Activation of L. plantarum A50

*Lactiplantibacillus plantarum* A50 has been deposited at the Key Laboratory of Forage Cultivation, Processing, and Highly Efficient Utilization, Ministry of Agriculture, PR of China. The raw data have been uploaded to the NCBI GenBank database, with the accession number SUB 13987624. Upon removal from an ultra-low-temperature refrigerator set at $-80\ ^{\circ}\mathrm{C}$, *L. plantarum* A50 was naturally thawed and subsequently incubated without agitation on MRS solid medium at $37\ ^{\circ}\mathrm{C}$ for 12 h. The inoculum of *L. plantarum* A50 was obtained after three generations of activation.

### 2.2. Research on the Growth Characteristics and Tolerance of L. plantarum A50

*L. plantarum* A50 was introduced into MRS broth, while blank MRS broth was used as a reference. The incubation of both samples occurred in a constant-temperature incubator at $37\ ^{\circ}\mathrm{C}$ for 24 h. The MRS broth's pH value and optical density at 600 nm (OD600) value were determined every 2 h to create growth and acid production curves. The MRS broths were prepared with various pH values using lactic acid (Macklin Biochemical Technology Co., Ltd., Shanghai, China) and sodium lactate (Macklin Biochemical Technology Co., Ltd., Shanghai, China), while different salt concentrations were achieved by incorporating NaCl (Macklin Biochemical Technology Co., Ltd., Shanghai, China) into the MRS broths. *L. plantarum* A50 was inoculated into MRS broths with varying pH levels and salt concentrations. After incubation for 12 h at $37\ ^{\circ}\mathrm{C}$ in a controlled environment, OD600 measurements were obtained to assess the acid, alkaline, and salt resistance of *L. plantarum* A50. Based on the research conducted by Wang et al. [29], we classified OD values within the range of 1.0 to 1.5 as indicative of "normal growth," OD values ranging from 0.5 to 1.0 as "weak growth," and OD values below 0.5 as indicating "no growth." The ability of *L. plantarum* A50 to utilize carbohydrates from 49 different compounds was investigated using the Api 50ch kit (Biomacrieux, Marcy l' Etoile) [30]. The OD600 measurements obtained in this study were determined using a spectrophotometer, while pH values were determined through the use of an electrode pH meter (PHS-3C, INESA Scientific Instrument Co., Ltd., Shanghai, China).

### 2.3. Research on the Ability of L. plantarum A50 Metabolites to Degrade Nitrite

2.3.1. The Capacity of *L. plantarum* A50 to Degrade Nitrite

The MRS broth was prepared by adding precise quantities of $NaO_2$ (Macklin Biochemical Technology Co., Ltd., Shanghai, China) to the MRS broth, achieving a concentration of 100 mg/L of nitrite. Subsequently, *L. plantarum* A50 was inoculated into the MRS broth and placed in a constant-temperature incubator at $37\ ^{\circ}\mathrm{C}$. The nitrite content was then assessed at regular intervals of 2 h to construct a nitrite degradation curve. Throughout this study,

the Griess method [31] was employed to quantify nitrite concentrations, with absorbance being measured at a wavelength of 538 nm.

### 2.3.2. The Influence of Varying Acidity on the Degradation of Nitrite

By utilizing lactic acid and sodium lactate, the pH level of deionized water was adjusted to 3, 3.5, 4, 4.5, 5, 5.5, and 6. Subsequently, $NaO_2$ was added to deionized water at different pH levels to configure a nitrite solution with a concentration of 200 mg/L. The samples were then incubated in a constant-temperature chamber at 37 °C for 48 h, with regular assessments of the nitrite salt levels every 12 h. Lastly, the decomposition of nitrite salts under various pH conditions was observed.

### 2.3.3. The Capacity of Metabolites Derived from *L. plantarum* A50 and MRS Broth in the Degradation of Nitrite

*L. plantarum* A50 was introduced into MRS broth and incubated at 37 °C in a thermo-static incubator for 48 h, with MRS broth serving as the control. Following the incubation period, both the *L. plantarum* A50 fermentation broth and the MRS broth without the added strain were subjected to centrifugation (4 °C, 10,000 g/min, 15 min). The resulting super-natants were then filtered through a 0.22 μm sterile membrane filter (Millipore Corporation, NYSE, USA) to obtain the *L. plantarum* A50 fermentation broth supernatant and the MRS broth supernatant, respectively. At this time, the pH of the *L. plantarum* A50 fermentation broth and the MRS broth supernatant measured 3.46 and 5.52, respectively. The pH of a portion of the MRS broth supernatant was adjusted to 3.46, matching the pH of the *L. plantarum* A50 fermentation supernatant. $NaO_2$ was introduced into both the *L. plantarum* A50 fermentation broth supernatant (at pH 3.46) and the MRS broth supernatant (at pH 3.46 and 5.52) to achieve a nitrite concentration of 130 mg/L. The incubation was conducted in a constant-temperature incubator at 37 °C for 24 h, during which the nitrite levels were measured every 6 h to assess the differences in nitrite degradation by the metabolites of *L. plantarum* A50 and MRS broth during the cultivation process.

### 2.3.4. The Influence of Varying Levels of Acidity on the Capacity of *L. plantarum* A50 Metabolites to Degrade Nitrite

The concentration of nitrite in the supernatant was adjusted to 140 mg/L by adding an appropriate amount of $NaO_2$. A portion of the supernatant from *L. plantarum* A50 was then subjected to heat treatment in a water bath at 100 °C for 15 min to inactivate the enzymes present. At this point, the pH of the enzyme-inactivated supernatant was measured to be 3.38. Subsequently, the pH of the uninactivated supernatant was adjusted to 3.38, while the pH of the enzyme-inactivated supernatant was adjusted to 3.88, 4.38, 4.88, and 5.38 using lactic acid and sodium lactate. The supernatant was then incubated at a constant temperature of 37 °C for 24 h, with measurements of nitrite content taken every 6 h. The objective of this experiment was to investigate the capability of A50 metabolites to eradicate nitrite in varying pH conditions.

### 2.4. Conducting an Untargeted Metabolomics Analysis on the Fermentation Broth of *L. plantarum* A50

#### 2.4.1. Sample Preparation

*L. plantarum* A50 and MRS broth underwent incubation at a precisely controlled temperature of 37 °C for a duration of 36 h. Following this, the fermentation broth was subjected to vigorous centrifugation (4 °C, 8000 g/min, 15 min) to obtain the supernatant. The supernatant was then filtrated through a sterile 0.22 μm filter and promptly stored in an ultra-low-temperature refrigerator (−80 °C).

Upon retrieval, the samples were thawed on a bed of ice, and 100 μL of each sample was carefully transferred into sterile and enzyme-free 2 mL centrifuge tubes. To each tube, 400 μL of meticulously chilled methanol (−20 °C) was added, ensuring thorough mixture by vortexing for 60 s. The samples were then subjected to another round of swift centrifugation (4 °C, 10,000 g/min, 10 min), and the resulting supernatant was vacuum-concentrated

and freeze-dried. To prepare the samples for LC-MS detection, precise amounts of 2-chlorophenylalanine (4 ppm) and phenylalanine (4 ppm) were added, concluding with a thorough filtration through sterile 0.22 μm filters. Subsequently, a comprehensive mixing was performed. To assess the reproducibility of the entire analytical process, an equal volume of metabolites from all samples was collected and combined to form a quality control (QC) sample.

### 2.4.2. LC-MS/MS Data Analysis

The samples were subjected to LC-MS/MS analysis utilizing a Thermo UHPLC-Q Exactive HF-X system, which was equipped with an ACQUITY HSS T3 column (100 mm × 2.1 mm i.d., 1.8 μm; Waters, MA, USA) at Majorbio Bio-Pharm Technology Co. Ltd. (Shanghai, China). The mobile phases comprised a mixture of 0.1% formic acid in water and acetonitrile (solvent A), with a ratio of 95:5 (*v/v*), and a combination of 0.1% formic acid in acetonitrile, isopropanol, and water (solvent B), with a ratio of 47.5:47.5 (*v/v*). For positive ion mode separation, the gradient was as follows: from 0% to 20% of mobile phase B within 0–3 min, followed by an increase from 20% to 35% within 3–4.5 min, and subsequently, a rise from 35% to 100% within 4.5–5 min. Mobile phase B was maintained at 100% for 5–6.3 min, and then decreased from 100% to 0% within 6.3–6.4 min. Finally, mobile phase B was kept at 0% for 6.4–8 min. As for negative ion mode separation, the gradient was as follows: an increase in mobile phase B from 0% to 5% within 0–1.5 min, followed by a rise from 5% to 10% within 1.5–2 min, and subsequently, an increase from 10% to 30% within 2–4.5 min. During the 4.5–5 min interval, mobile phase B was elevated from 30% to 100%, and maintained at 100% from 5–6.3 min. The decrease in mobile phase B from 100% to 0% occurred within 6.3–6.4 min, and mobile phase B was maintained at 0% from 6.4–8 min. The flow rate was set at 0.40 mL/min, and the column temperature was maintained at 40 °C. The mass spectrometric analysis was conducted using a Thermo UHPLC-Q Exactive HF-X mass spectrometer, equipped with an electrospray ionization (ESI) source. The analysis was performed in both positive and negative modes. The optimized parameters included a source temperature of 425 °C, a sheath gas flow rate of 50 arb, an auxiliary gas flow rate of 13 arb, and an ion-spray voltage floating (ISVF) of −3500 V in negative mode and 3500 V in positive mode, respectively. The collision energy for MS/MS was varied in a rolling fashion, employing 20–40-60 V. The full MS resolution was set at 60,000 and the MS/MS resolution at 7500. Data acquisition was carried out utilizing the data-dependent acquisition (DDA) mode, with a mass range of 70–1050 $m/z$.

### 2.4.3. Data Analysis

The LC/MS raw data underwent preliminary processing using Progenesis QI software provided by Waters Corporation, located in Milford, KS, USA. Subsequently, a CSV-formatted three-dimensional data matrix was exported, containing crucial information such as sample details, metabolite names, and intensity of mass spectral response. The data matrix was carefully curated by eliminating internal standard peaks and any anticipated false positive peaks, encompassing noise, column bleed, and derivatized reagent peaks. Simultaneously, metabolite identification was executed via meticulous exploration of renowned databases, primarily focusing on the HMDB (http://www.hmdb.ca/) and Metlin (https://metlin.scripps.edu/) accessed on 15 October 2023.

### 2.5. Statistical Analysis

The experiments were performed with a total of 6 repetitions. The preprocessed data matrices were subjected to a technique called orthogonal least partial squares discriminant analysis (OPLS-DA), utilizing the ropls package in R (Version 1.6.2). To identify metabolites that exhibited significant differences, variable weight values (VIP) derived from the OPLS-DA model and *p*-values from Student's *t*-test were used. Metabolites with VIP values exceeding 1 and *p*-values less than 0.05 were considered statistically significant. The metabolic pathways associated with these differentially expressed metabolites

were determined through the process of metabolic pathway annotation, employing the KEGG database (https://www.kegg.jp/kegg/pathway.html) accessed on 20 October 2023. Pathway enrichment analysis was conducted using the scipy.stats Python package (1.6.0). Visual representation of the data was achieved by generating line plots using Origin 2022, while bar plots were created using GraphPad Prism 10.

## 3. Results

### 3.1. The Growth Trajectories and Physiological as well as Biochemical Markers of Lactobacillus plantarum A50

The growth rate profile of *L. plantarum* A50 is depicted in Figure 1 whereby a higher optical density (OD) indicates a greater pace of growth for the lactic acid bacteria. In the initial 0–4 h of incubation, the growth of *L. plantarum* A50 exhibited a relatively sluggish pattern, which subsequently accelerated during the 6–16 h interval. Notably, beyond the 16 h mark, the growth rate displayed a gradual incline. Moving on to the acid production capacity, Figure 1 visualizes the curve while considering an initial pH value of 5.61 at the beginning of incubation. During the initial 0–6 h of incubation, the acid production rate remained subdued but significantly escalated from 8–12 h. Moreover, the pH value registered a gradual decrease after the 12 h mark, reaching its nadir at 3.66 after 24 h.

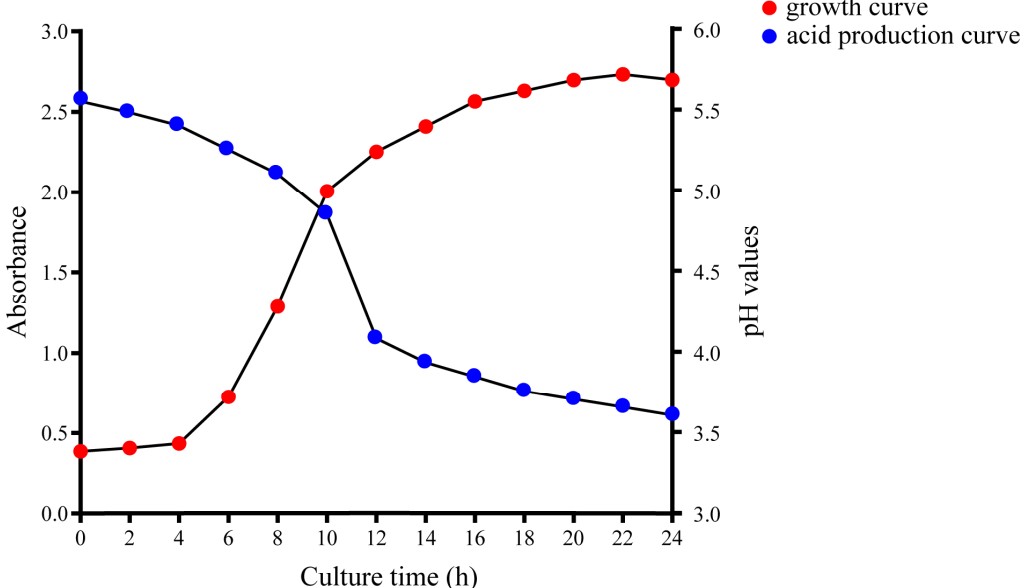

**Figure 1.** The growth curve and acid production curve of *L. plantarum* A50.

### 3.2. Research on the Nitrite Degradation Ability of L. plantarum A50

The degradation progress of nitrite by *L. plantarum* A50 over 24 h is illustrated in Figure 2A. The initial nitrite degradation rate by *L. plantarum* A50 was relatively slow within the first 6 h. However, a notable acceleration in degradation was observed during the subsequent 6–12 h, which yielded the highest degradation rate. At the 12 h mark, the degradation rate of nitrite reached an impressive 82.67%. During the 12–24 h period of incubation, the degradation rate of nitrite by *L. plantarum* A50 exhibited a gradual and steady decline. Remarkably, an extraordinary degradation rate of 99.06% was attained at 24 h.

The degradation trends of nitrite under varying pH levels are depicted in Figure 2B. As the pH decreased, the extent of nitrite degradation exhibited an increment. Within the pH range of 4.5–6.0, the nitrite concentration in the solution experienced minimal changes over time, showing a degradation rate ranging from 10.53% to 17.38% after 48 h. Remarkably, at pH 4, the nitrite content declined significantly, resulting in a degradation rate of 51.60% within the same time frame. Notably, at pH 3 and 3.5, the nitrite degradation rates were exceptionally high, reaching 95.00% and 83.21%, respectively.

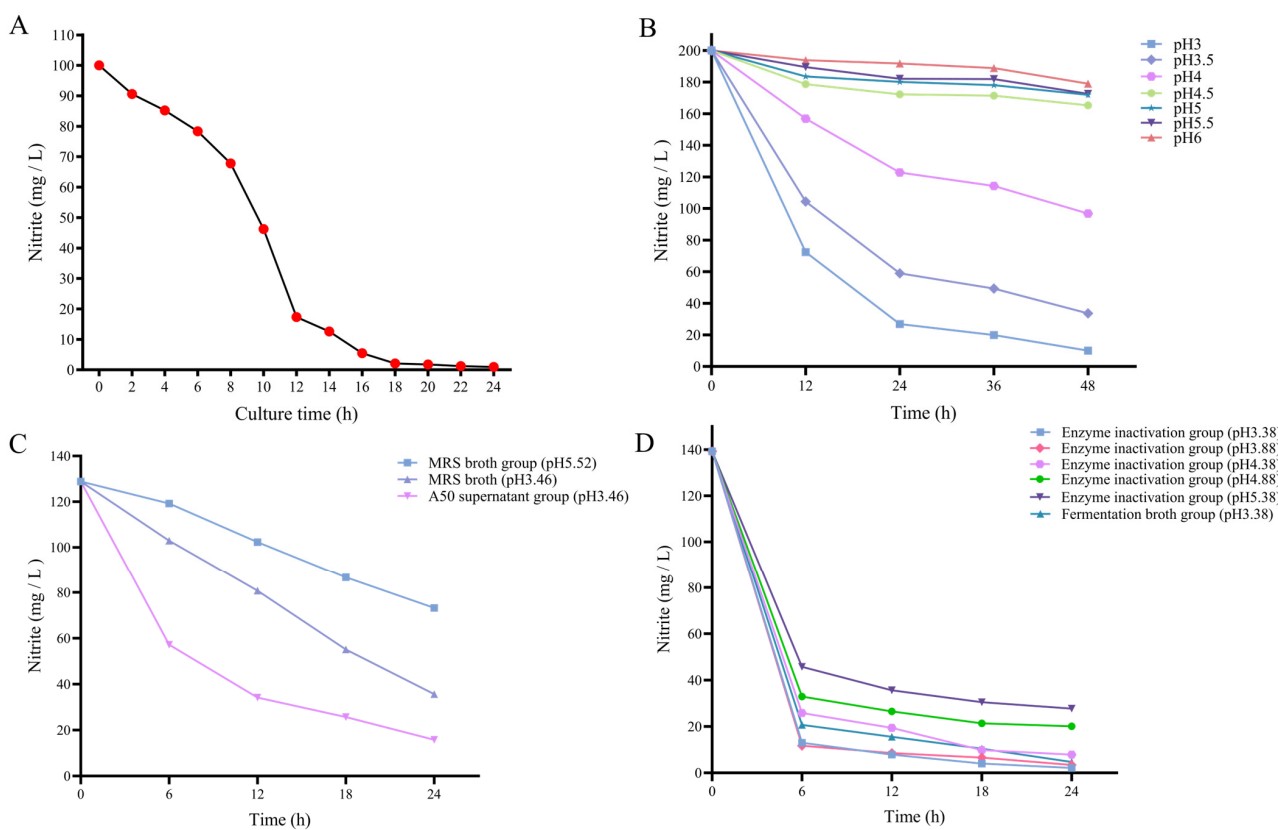

**Figure 2.** Degradation profiles of nitrite by *L. plantarum A50* (**A**); Nitrite degradation profiles at various pH levels (**B**); Nitrite degradation profiles by metabolites of *L. plantarum* A50 and MRS broth (**C**); Nitrite degradation profiles by metabolites of *L. plantarum* A50 under different pH conditions (**D**).

The capacity of metabolites from *L. plantarum* A50 and MRS broth to degrade nitrite is illustrated in Figure 2C. Among the experimental groups, the control MRS broth (pH 5.62) exhibited the least effective nitrite degradation, whereas the *L. plantarum* A50 supernatant group showcased the most remarkable nitrite degradation. Remarkably, the *L. plantarum* A50 supernatant group displayed the swiftest rate of nitrite degradation within the initial 12 h, achieving an impressive degradation rate of 73.48%. Subsequently, there was a gradual decline in the nitrite content in the *L. plantarum* A50 supernatant group, with degradation rates reaching 80.04% and 87.72% at 18 h and 24 h, respectively.

As depicted in Figure 2D, the degradation profile of *L. plantarum* A50 metabolites illustrates that the nitrite degradation rate of *L. plantarum* A50 metabolites was most rapid in the initial 6 h. Specifically, at pH 3.38, the nitrite degradation rate was 90.66% in the *L. plantarum* A50 fermentation broth, while the enzyme inactivation group exhibited rates of 91.58%, 81.45%, 76.39%, and 67.18% at pH 3.88, 4.38, 4.88, and 5.38, respectively. Notably, the nitrite content was minimized after 24 h of incubation, with the enzyme-inactivated group displaying degradation rates of 98.48%, 97.56%, 94.34%, 84.86%, and 80.12% at pH values of 3.38, 3.88, 4.38, 4.88, and 5.38, respectively.

### 3.3. Metabolite Analysis of L. plantarum A50

The fermentation broths of *L. plantarum* A50 and MRS were subjected to non-targeted LC-MS analysis. To ensure accurate identification of differential metabolites, quality control (QC) procedures were implemented to minimize errors. The QC-treated samples exhibited improved aggregation, with approximately 70% demonstrating an RSD of less than 30%, ensuring the reliability of the data [32]. Figure 3 displays the PLS-DA score plot, which validates the metabolite differences between the two groups and shows the multivariate analysis. This plot is commonly used to visualize the classification effect of the model,

where a greater separation between the sample groups indicates a more significant classification effect. Furthermore, the data points within the QC cohort exhibited a remarkable convergence, implying a commendable level of reproducibility during the procedure of sample acquisition. For the analysis of cations, the utilization of intrasample group clustering paired with intergroup discretization yielded an R2X value of 0.764, R2Y value of 0.99, and Q value of 0.954. As for the analysis of anions, the application of intrasample group clustering in combination with intergroup discretization produced an R2X value of 0.773, R2Y value of 0.98, and Q value of 0.956. Furthermore, to evaluate the potential issue of overfitting in the model, PLS-DA alignment plots were employed. It is worth noting that in cation and anion modes, *L. plantarum* A50 and MRS broth were distinctly segregated into two groups, underscoring the influential role of *L. plantarum* A50 on the metabolites present in MRS broth.

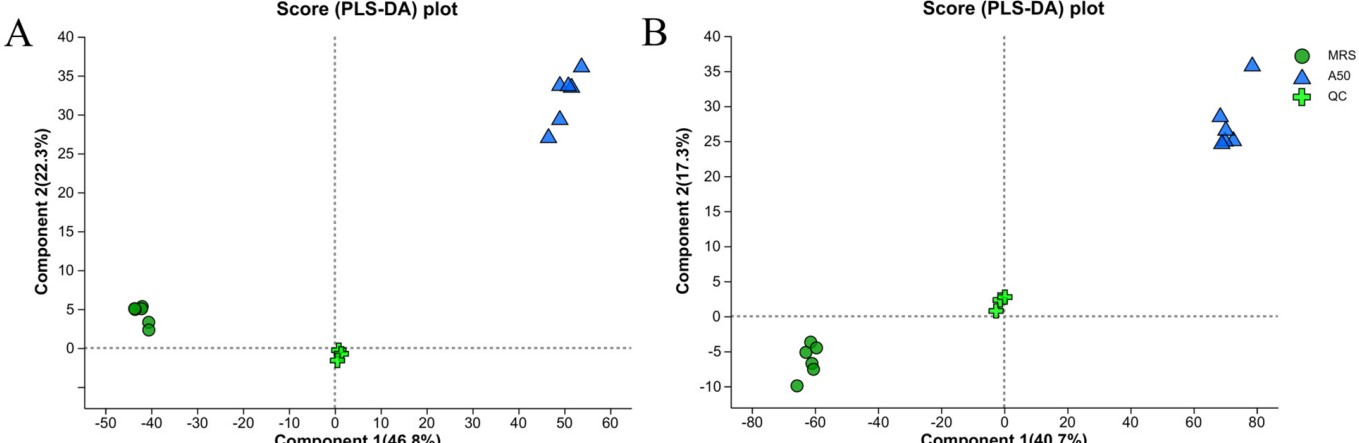

**Figure 3.** The cation (**A**) and anion (**B**) OPLS−DA score plot of *L. plantarum* A50 and MRS broth fermented solution.

The fermented solutions of *L. plantarum* A50 and MRS broth were used as the experimental group and control group, respectively. Following Figure 4, a total of 362 different metabolites were identified in both the experimental and control groups, and these were subsequently classified based on their chemical structures. These classifications included 91 lipids and lipid-like molecules, 85 organic acids and derivatives, 47 organoheterocyclic compounds, 35 organic oxygen compounds, 23 phenylpropanoids and polyketides, 19 benzenoids, 6 alkaloids and derivatives, 6 nucleosides, nucleotides, and analogues, 2 organosulfur compounds, 1 organic nitrogen compound, 1 organic polymer, and 41 unclassified compounds. According to Figure 5A, 52 cationic substances displayed an up-regulation trend, whereas 99 cationic substances displayed a down-regulation trend. Moreover, 81 anionic substances exhibited an up-regulation pattern, while 130 anionic substances showed a down-regulation pattern.

Figure 6 presents the annotated differential metabolites from the HMDB, categorized under the superclass hierarchy. Lipids and lipid-like molecules, organic acids and derivatives, organic oxygen compounds, and organoheterocyclic compounds exhibited a higher percentage of differential metabolites. Therefore, these four superclass hierarchies were selected for visualization and mapping purposes in this study.

As depicted in Figure 6A, a majority of the metabolites in the lipids and lipid-like molecules superclass displayed down-regulation. Examples include 9S-hydroxy-11,15-dioxo-5Z,13E-prostadienoic acid, 14-HDOHE, vulgarin, ginsenoside C-K, indosterol, (+)-dehydrovomifoliol, progestoral, testosterone undecanoate, and 6-Epi-7-isocucurbic acid glucoside. Figure 6B illustrates that the organic acids and derivatives superclass primarily consists of amino acids, amides, peptides, and their derivatives. Notable examples include hydroxypropionic acid, histidylprolineamide, serylasparagine, N-linoleoyl methionine, Leu-Arg-Asn-Arg, N-linoleoyl glutamine, L-methionine, 2-acetylornithine,

isoleucyl-aspartate, spermic acid 2, N-docosahexaenoyl glutamine, and isoleucyl-lysine. Moreover, Figure 6C depicts that the organoheterocyclic compounds superclass primarily consists of 5-methylcytosine, cimetidine, 2-beta-hydroxymedroxyprogesterone, dipropyl-5-CT, 5-methyl-1H-pyrazole-3-carboxylic acid, 2-(2-phenylimidazo [1,2-a]pyridin-8-yl) acetamide, navoximod, hypoxanthine, and 5-methylcytosine. Lastly, Figure 6D shows that the organic oxygen compounds superclass mainly comprises glucose and galactose derivatives such as gluconolactone, tyramine glucuronide, pyridine N-oxide glucuronide, O-desmethyltramadol glucuronide, 6″-O alpha-D-galactopyranosylciceritol, and 2-deoxy-2-fluoro-alpha-D-galactopyranose.

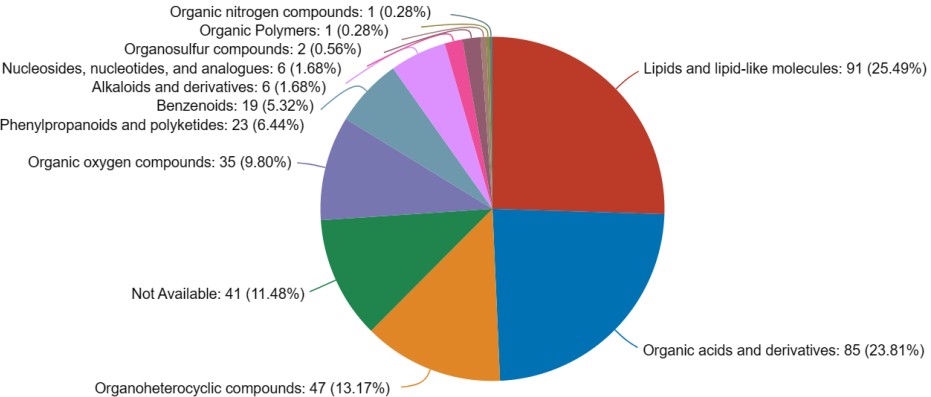

**Figure 4.** Classification diagram of metabolite chemical structures for *L. plantarum* A50 and MRS broth.

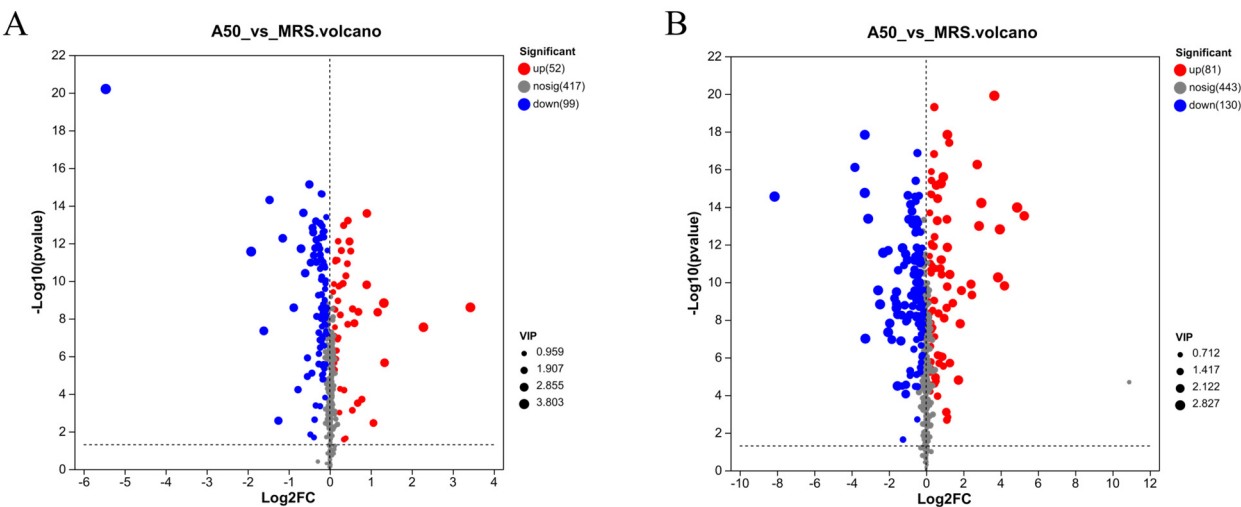

**Figure 5.** Differential metabolite volcano plot of *L. plantarum* A50 and MRS broth: Cations (**A**) and Anions (**B**).

To explore the modified metabolic pathways and their biological significance, KEGG pathway enrichment analysis was performed to identify the biological pathways involved in differentially expressed metabolites and their respective roles (Figure 7A,B). The results suggest that nine metabolic pathways were significantly impacted with lower *p*-values and higher pathway impact factors observed between *L. plantarum* A50 and the control group. These pathways include those associated with glycerophospholipid metabolism, ABC transporters, nucleotide metabolism, arginine biosynthesis, pyrimidine metabolism, purine metabolism, aminoacyl-tRNA biosynthesis, pathogenic *Escherichia coli* infection, and D-amino-acid-metabolism-related pathways.

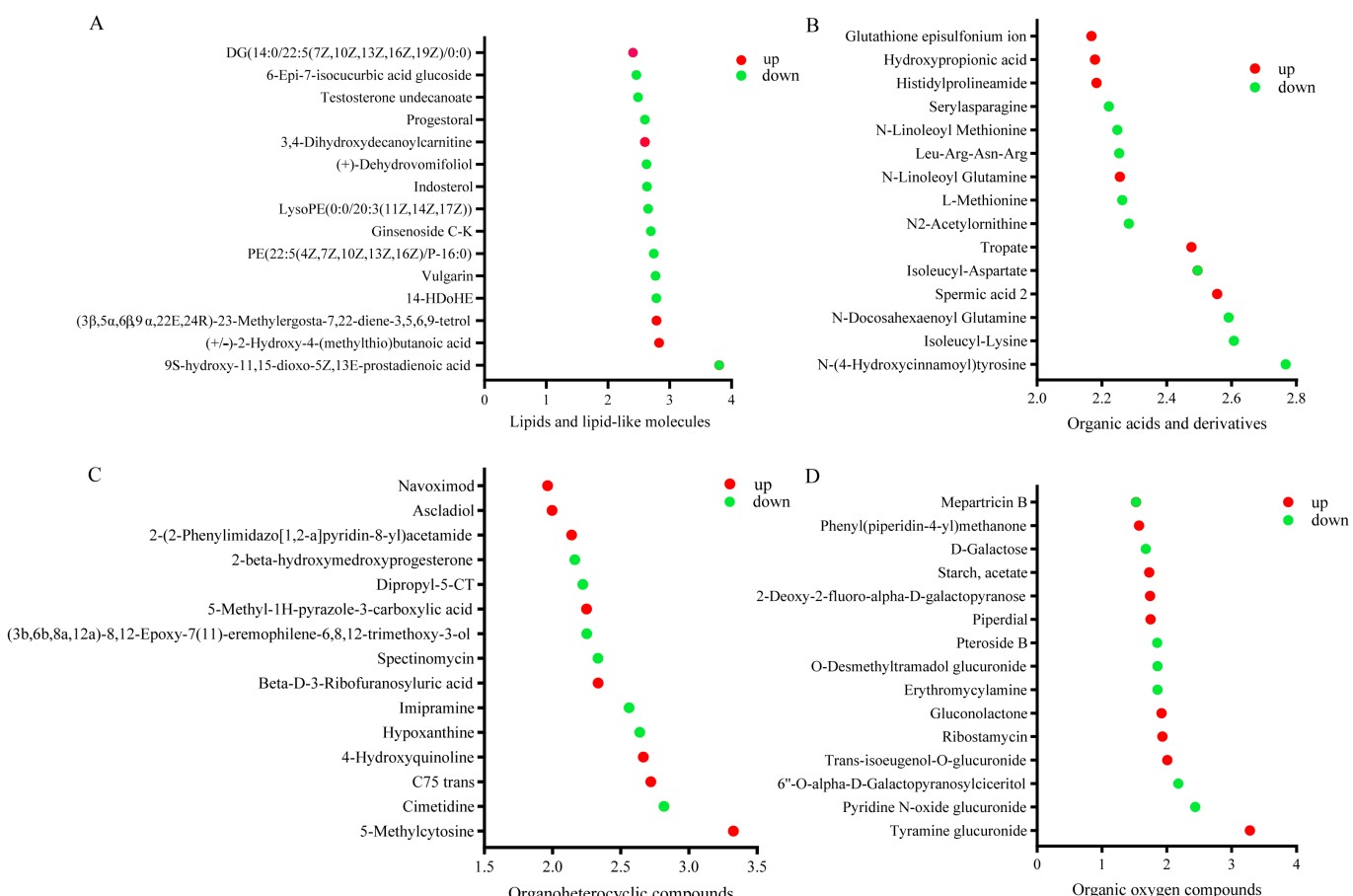

**Figure 6.** The VIP value plot displaying differential metabolites among Lipids and lipid-like molecules (**A**), Organic acids and derivatives (**B**), Organoheterocyclic compounds (**C**), and Organic oxygen compounds (**D**).

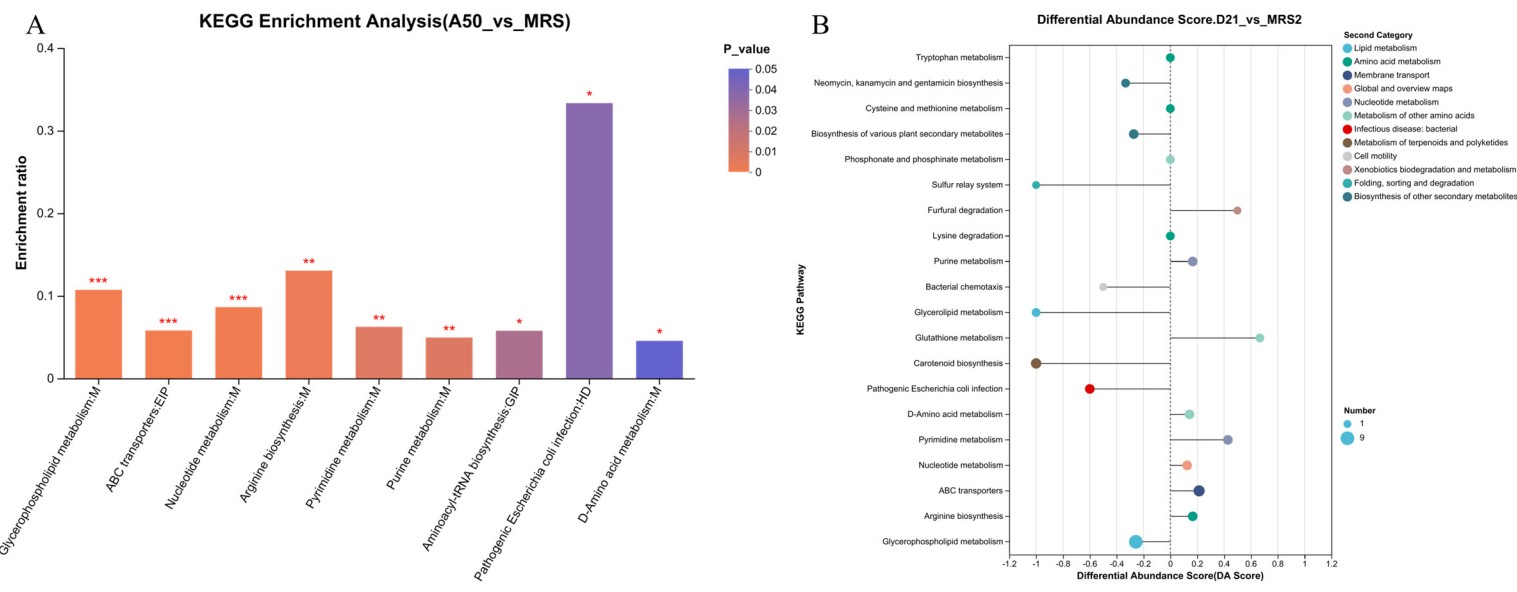

**Figure 7.** KEGG enrichment analysis plot (**A**) and KEGG pathway differential abundance score plot (**B**). *, $0.01 < p < 0.05$; **, $0.001 < p < 0.01$; ***, $p < 0.001$.

## 4. Discussion

### 4.1. Physiological Characteristics of L. plantarum A50

The findings of this study indicate that *L. plantarum* A50, isolated from silage, exhibits a high acid-producing ability, with a fermentation liquid pH of 3.66 after 24 h of cultivation. *L. plantarum* A50 demonstrates good growth at pH values ranging from 4 to 9 and salt concentrations of 5%, demonstrating its excellent salt, acid, and alkali tolerance, positioning it as a desirable lactic acid bacterium for fermentation (Supplementary Figure S1). In comparison to traditional carbohydrate fermentation tests, the Api 50ch method proves to be a faster and more accurate approach [33]. Consistent with the previous research by McDonald et al. [34], it is observed in this study that *L. plantarum* A50 shows excellent utilization of the 24 tested carbohydrates for fermentation (Supplementary Table S1). However, the fermentation substrates of *L. plantarum* A50 differ from those screened by Cai et al. [35] for *Pediococcus pentosaceus*. This discrepancy may be attributed to variations in bacterial species or geographical environments [36].

### 4.2. Degradation of Nitrite by L. plantarum A50

In the present study on the degradation of nitrite in acidic conditions, it was observed that the nitrite content decreased as the pH decreased. Interestingly, the rate of nitrite degradation was found to increase significantly when the pH dropped below 4. This suggests that acid degradation gradually becomes the dominant factor in nitrite degradation when the pH is below 4, which aligns with earlier research [37]. In our study, we noticed that the degradation of nitrite by *L. plantarum* A50 mainly occurred between 6 and 12 h of incubation, with a pH of 4.14 at the 12 h mark. Therefore, it can be inferred that the degradation of nitrite by *L. plantarum* A50 may not be primarily driven by acid degradation. However, after 12 h of incubation, the pH fell below 4.0, which implies that the degradation of nitrite may then be dominated by acid degradation. Consequently, we conducted further investigation into the mechanism of nitrite degradation by *L. plantarum* A50 metabolites.

The metabolites derived from the strain *L. plantarum* A50 exhibited a remarkable nitrite degradation rate of 87.72% after 24 h of incubation. This result indicates that the metabolites produced by *L. plantarum* A50 possess a strong ability to scavenge nitrite. In parallel, the control group, consisting of MRS broth, also demonstrated nitrite degradation capability. Interestingly, it was observed that MRS broth and acid exhibited a synergistic effect in nitrite degradation, which is in line with the findings from Xia et al. [38]. Specifically, Xia et al. [38] proposed that MRS broth contains components such as peptone and yeast extracts, which may undergo compositional changes following autoclave treatment, thereby contributing to the reduction of nitrite. These results provide further insights into the potential mechanisms underlying nitrite degradation facilitated by the metabolites of *L. plantarum* A50 and the influence of MRS broth in this process.

Despite the inactivation of enzymes in the supernatant of *L. plantarum* A50, its strong ability to degrade nitrate can still be observed. *L. plantarum* A50 demonstrated maximized degradation efficacy of its metabolites at a pH of 3.38, achieving an impressive degradation rate of 98.48%. Even when the pH rose to 5.38, *L. plantarum* A50 still exhibited notable degradation capacity with a rate of 80.20%. These findings unequivocally indicate that the degradation of nitrite by *L. plantarum* A50 metabolites progressively intensified as the pH decreased, thus confirming the collaborative influence of metabolites and acidity in nitrite degradation.

### 4.3. Metabolite Analysis of L. plantarum A50

Lipids and lipid-like metabolites were predominantly down-regulated, indicating the utilization of lipids by *L. plantarum* A50 for metabolic processes. Lactic acid bacteria can convert fatty acids into energy through lipid metabolism [39]. Fatty acids play a crucial role in providing energy to lactic acid bacteria and facilitating the maintenance of their required energy supply for vital life activities [40]. Additionally, lipids serve as important components of cell membranes, and lactic acid bacteria synthesize phospholipids

essential for maintaining the integrity and functionality of these membranes through lipid metabolism [41]. Signaling molecules produced by lactic acid bacteria as a result of lipid metabolism, such as fatty acid metabolites and derivatives, are implicated in intercellular signaling and regulation. These signaling molecules can impact cell growth, metabolism, and physiological functions [42]. Furthermore, lactic acid bacteria lipid metabolites form a protective lipid barrier on the cell membrane, enabling survival in unfavorable environments characterized by higher acidity and anaerobic conditions as well as protecting cells from oxidative stress and damage [43]. Huang et al. [44] observed that *Lactobacillus plantarum* 9010 cells without nitrite supplementation possessed a smooth surface and intact cell shape. However, as the nitrite concentration increased, some *L. plantarum* 9010 cells displayed surface irregularities, and with further increases in nitrite concentration, certain *L. plantarum* cells experienced rupture, resulting in cytoplasmic leakage. Previous studies have demonstrated that nitrite causes severe damage to bacterial DNA [45]. Bacteria combat this damage by inducing an SOS response to accelerate replication and transcription, enabling the generation of more individuals as a response to environmental stress [45]. Therefore, it is plausible to propose that lipid metabolism may serve as a potential mechanism through which *L. plantarum* A50 protects cells from nitrite-induced stress and damage by effectively scavenging nitrite ions.

The classification of organic acids and derivatives primarily comprises amino acids, amides, peptides, and their derivatives, which play crucial physiological roles in organisms, involving biological processes like protein synthesis, hormone regulation, and cell signaling [46,47]. Research indicates that most amino acids possess the ability to degrade nitrite [48]. Ornithine acts as a precursor for proline and, thus, an enhanced metabolism of N2-acetylornithine may impact the production of proline and potentially promote the formation of histidylprolineamide. Amino acids can be metabolized through methionine, which in turn participates in sulfur metabolism [49]. Enzymatic reactions, such as hydroxypropionic acid hydroxylase, can convert hydroxypropionic acid into alanine. The transmembrane transport of both alanine and methionine is vital for cellular metabolism and nutrient transportation [50]. It has been discovered that alanine can stimulate microbial activity during the nitrogen cycle through cleavage [51]. Therefore, the heightened metabolic process of hydroxypropionic acid and L-methionine signifies an elevated level of metabolic activity within *L. plantarum* A50, thereby fostering the advancement of nitrite degradation [52]. Threonine and methionine serve as substrates for isoleucine synthesis, and their synthesis and catabolism under diverse developmental and environmental conditions also impact the availability of isoleucine [53]. Isoleucine participates in metabolic pathways and biological functions of cells, and it has a specific function in the protein-folding process of membrane proteins [54]. Isoleucyl-aspartate is a peptide that is formed by the combination of two amino acids, isoleucyl and aspartate, through peptide bonding. A previous study [55] discovered a correlation between nitrite reductase and acetolactate synthase activities in cells following the activities of glutamine synthetase, aspartate aminotransferase, and phosphoglycan dehydrogenase. Aspartate aminotransferase is closely associated with the metabolism of aspartic acid and is involved in the conversion of aspartic acid and $\alpha$-ketoglutaric acid [56]. Serylasparagine is a type of secondary metabolite that contains asparagine. Therefore, we hypothesized that isoleucyl-aspartate and serylasparagine may play a direct or indirect role in nitrite degradation. Bacteria can reduce aspartic acid to semialdehyde, which then condenses with pyruvate to form lysine. Lysine provides energy for biological metabolism and promotes glucose synthesis [57]. Thus, the enhancement of isoleucyl-lysine metabolism may contribute to nitrite reduction. Leu-Arg-Asn-Arg is a peptide that may mimic the functions of leucine, arginine, and asparagine. Therefore, Leu-Arg-Asn-Arg may have the ability to reduce nitrite. Spermic acid 2 belongs to the class of organic compounds known as $\beta$-amino acids and their derivatives. According to a literature review, there have been few published articles specifically addressing the role of spermic acid 2, and further research is needed in this regard. N-linoleoyl glutamine, N-docosahexaenoyl glutamine, and N-linoleoyl methionine are derivatives of fatty acids.

Increasing the concentration of fatty acids may activate defense systems and enhance tolerance to nitrite [58]. However, the exact mechanism by which fatty acids interact with nitrites remains unclear, warranting further investigation. A study involving blue-green algae [59] has suggested that promoting glutamine synthetase activity can reduce nitrate and nitrite levels, but there is limited research available on the specific effects of N-linoleoyl glutamine and N-docosahexaenoyl glutamine, necessitating further study.

Predominantly derived from glucose and galactose in the classification of organic oxygen compounds, gluconolactone, tyramine glucuronide, pyridine N-oxide glucuronide, and O-desmethyltramadol glucuronide are generated by *L. plantarum* A50 through the process of glucose metabolism to produce derivatives. Lactic acid bacteria are incapable of utilizing complex carbohydrates, such as cellulose, as an energy source, whereas glucose, being a simple sugar, assumes a pivotal role in the energy metabolism of lactic acid bacteria [60]. Lactic acid bacteria can effectively harness the potential energy and essential raw materials for growth from glucose through glycolysis and other metabolic pathways. Glucose fermentation carried out by lactic acid bacteria leads to the production of organic acids like lactic acid and acetic acid [61]. These organic acids serve as the primary products of lactic acid bacteria fermentation, contributing to the regulation of the acid–base balance and inhibition of the growth of other detrimental microorganisms [62]. Simultaneously, organic acids also serve as a significant means of nitrite degradation. Furthermore, Ma et al. [63] propose that glucose augments the effectiveness of nitrite reductase, thereby facilitating denitrification mediated by denitrifying and anaerobic ammonia-oxidizing bacteria. Consequently, the metabolic utilization of glucose by lactic acid bacteria emerges as a pivotal avenue to augment the decomposition of nitrite.

D-galactose is a type of galactose, and both 6''-O-alpha-D-galactopyranosylciceritol and 2-deoxy-2-fluoroalpha-D-galactopyranose are considered galactose derivatives. Similar to glucose, D-galactose is also an important monosaccharide. Lactic acid bacteria are capable of converting D-galactose into pyruvate and lactic acid while generating intermediate metabolites like lactaldehyde and gluconic acid. According to Shi et al. [52], the presence of $NaNO_2$ in the fermentation medium increases the energy required for cellular survival, consequently elevating the demand for glucose. Moreover, in the presence of $NaNO_2$, carbohydrate metabolism is enhanced, leading to improved energy supply and enhanced tolerance to osmotic stress. In a study conducted by Li et al. [64], it was demonstrated that sucrose and glucose, which are involved in the metabolism of D-galactose, starch, and sucrose, can provide energy for the growth and proliferation of lactic acid bacteria, resulting in intensified anaerobic metabolism and reduced nitrite levels.

The presence of organoheterocyclic compounds in the metabolites of *L. plantarum* A50 holds crucial implications for the biological significance of this strain, albeit with limited specific investigations in this domain. Organoheterocyclic compounds represent a class of compounds comprising carbon and other elements such as oxygen, nitrogen, and sulfur, exhibiting diverse biological activities within living organisms. The organoheterocyclic compounds investigated in this study primarily encompass pyrazoles, imidazoles, isoindoles, indazoles, and indolecarboxylic acids and their derivatives. Prior research has demonstrated the antibacterial and anti-inflammatory effects of pyrazoles [65], imidazoles [66], indazoles [67], isoindoles, and indolecarboxylic acids and derivatives [68,69]. These substances possess the ability to exert an impact on the intricate dynamics between lactic acid bacteria and their microbial counterparts, facilitating their sustenance and proliferation within a specific milieu through the suppression of growth and metabolic processes of the latter [70]. Furthermore, the presence of these compounds exhibits antioxidant effects; for instance, certain components of indolecarboxylic acids and derivatives may possess antioxidant activity that aids lactic acid bacteria in combating cellular oxidative stress [68]. This antioxidative capability assists in maintaining cellular homeostasis and function by scavenging free radicals and safeguarding essential intracellular molecules against oxidative damage. Hence, the down-regulation observed in this study of cimetidine, 2-beta-hydroxymedroxyprogesterone, and dipropyl-5-CT might have facilitated the prolif-

eration and reproductive capacity of *L. plantarum* A50. Conversely, the up-regulation of 5-methyl-1H-pyrazole-3-carboxylic acid, 2-(2-phenylimidazo[1,2-a]pyridine-8-yl)acetamide, and navoximod could potentially serve as a defense mechanism for *L. plantarum* A50 against external threats or aid in the production of bacteriostatic substances. The breakdown of hypoxanthine in organisms can be potentially perilous as it stimulates the production of urate and, most notably, hydrogen peroxide [71]. The decrease in the level of hypoxanthine observed in this study may suggest that *L. plantarum* A50 was reducing the generation of urate and hydrogen peroxide to minimize the oxidative harm inflicted upon cells and tissues. In bacteria, 5-methylcytosine primarily impacts gene expression and transcriptional processes, and it may assume a vital regulatory role in the SOS response [72]. Consequently, the up-regulation of 5-methylcytosine may indicate that *L. plantarum* A50 possesses the ability to actively counteract DNA damage and initiate the corresponding protective mechanisms.

Enrichment analysis using KEGG was performed to identify the biological pathways associated with differentially expressed metabolites and their roles in the survival, adaptation, and probiotic function of lactic acid bacteria. The metabolic pathways involved in glycerophospholipid metabolism, nucleotide metabolism, pyrimidine metabolism, and purine metabolism are essential for meeting the nutritional needs of lactic acid bacteria, providing important nutrients such as lipids, nucleotides, and amino acids [73–75]. Furthermore, glycerophospholipid metabolism contributes to the formation and structural integrity of lactic acid bacterial cell membranes, including the transport, signaling, and stability of phospholipid molecules [73]. In addition, lactic acid bacteria employ ABC transporter proteins to regulate the uptake and excretion of substances, maintain the balance between intra- and extracellular environments, and respond to environmental changes [76]. Additionally, metabolic pathways such as arginine biosynthesis [77], aminoacyl-tRNA biosynthesis [78], and D-amino acid metabolism [79] are involved in the synthesis and alteration of amino acids and proteins. These processes have an impact on intracellular enzyme activity, protein structure, and function, as well as the synthesis and release of metabolic products in lactic acid bacteria. D-amino acid is derived from cell wall synthesis, protein degradation, and external sources, and it plays a crucial role in the growth and metabolism of lactic acid bacterial cells [80]. Nonetheless, Soutourina et al. [81] suggest that an overabundance of D-amino acids can hinder bacterial growth by potentially hindering protein synthesis and resulting in an accumulation of tRNAs that do not match with D-amino acids. Aminoacyl-tRNA biosynthesis is employed by lactic acid bacteria to facilitate protein synthesis. This metabolic pathway involves the activation and binding of amino acids to tRNAs, providing the essential substrates and catalysts needed for the synthesis of proteins with specific amino acid sequences in lactic acid bacterial cells [82]. Apart from their catalytic role in protein synthesis, aminoacyl-tRNA synthetases can be regarded as fundamental signaling molecules. The ability of these enzymes to interact with tRNAs extends to other cellular nucleic acid molecules, thus regulating various biological processes including transcription, splicing, and translation [83]. The noticeable reduction of metabolites in pathogenic *Escherichia coli* infections suggests that lactic acid bacteria enhance resistance against harmful microorganisms, reinforce the immune response, and lower the risk of pathogenic-microbe-induced diseases [84]. In aggregate, these metabolic pathways serve vital functions in Lactobacilli, bolstering their proliferation and metabolic requirements, while also influencing the composition of their cell membranes, facilitation of material transportation, synthesis and modification of proteins, as well as enhancing resistance against pathogenic microorganisms.

## 5. Conclusions

*L. plantarum* A50 exhibits commendable tolerance to salt, acid, and alkali, allowing for fermentation with a diverse range of carbon sources. Moreover, it demonstrates an impressive capacity for the removal of nitrite, with a remarkable degradation rate of 99.06% within 24 h. The degradation of nitrite by *L. plantarum* A50 predominantly occurs in a non-acidic manner from 6–12 h, transitioning to an acidic process thereafter. Furthermore, the metabo-

lites produced by *L. plantarum* A50 show superiority in terms of nitrite-scavenging ability and exhibit synergistic effects when combined with acid. A comprehensive LC-MS-based untargeted metabolomic analysis of *L. plantarum* A50 metabolites yielded the identification of 342 metabolites that exhibited significant differences. The heightened metabolic levels of lipids and lipid-like molecules, organic acids and derivatives, organic oxygen compounds, and organoheterocyclic compounds, amongst other substances, may potentially contribute to the clearance process of nitrites. Notably, upon conducting KEGG differential metabolic pathway enrichment analysis, nine distinct differential metabolic pathways were identified. These pathways are proposed to provide essential nutrients to support the growth of *L. plantarum* A50, maintain the integrity of cellular structure and function, facilitate substance transport, regulate metabolic activities, and enhance resilience against pathogenic microorganisms. Despite the insightful findings derived from this single-omics investigation, it is crucial to acknowledge its inherent limitations. Therefore, future studies should aim to delve further into the mechanism of nitrite degradation by *L. plantarum* A50 by employing advanced methodologies such as genomics and transcriptomics.

**Supplementary Materials:** The following supporting information can be downloaded at: https://www.mdpi.com/article/10.3390/fermentation10020092/s1, Figure S1: The ability of *L. plantarum* A50 to withstand acidity and alkalinity (A) as well as salt tolerance (B); Table S1: The carbohydrate fermentation characteristics of *L. plantarum* A50.

**Author Contributions:** Conceptualization, methodology, data curation, writing—original draft preparation, and writing—review and editing: J.A. Writing—original draft preparation, writing—review and editing: L.S. Methodology: M.L. Writing—review and editing: R.D. Writing—review and editing: Q.S. Investigation and resources: Z.W. Writing—review and editing: G.G. Project administration and funding acquisition: Y.J. All authors have read and agreed to the published version of the manuscript.

**Funding:** This work was supported by Special Fund for the Commercialization of Scientific and Technological Achievements Project in Inner Mongolia (2021CG0006) and National Dairy Technology Innovation Center (2021-National Dairy Innovation Center-1), China.

**Institutional Review Board Statement:** Not applicable.

**Informed Consent Statement:** Not applicable.

**Data Availability Statement:** Sequencing data for 16S rDNA gene sequence were stored in NCBI with GenBank accession number SUB 13987624.

**Acknowledgments:** We thank Yihe Lvjin Agricultural Development Co., Ltd. (Aluhorqin Banner, Chifeng, China) for providing alfalfa and Inner Mongolia Agricultural University (Hohhot, China) for providing the microbiology laboratory.

**Conflicts of Interest:** The authors declare no conflicts of interest.

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
