# Peer review of "Mechanistic Insights into Nitrite Degradation by Metabolites of L. plantarum A50: An LC-MS-Based Untargeted Metabolomics Analysis"

_fermentation, doi:10.3390/fermentation10020092_

Round 1

Reviewer 1 Report

Comments and Suggestions for Authors

Thank you for your manuscript.

While I appreciate the efforts of the work, I recommend revising the manuscript to strengthen the research appeal. 

1. The study area and methods are quite overlapped with previous studies such as Huang et al (LWT, 2021), Yao et al (IFST, 2020), Li et al (Food Res Int, 2022). What is the difference between the results of this study and those of these studies?

2. Line 58: "lactic"? not "Lactic"

3. should re-write 2.3.2

4. Line 194: should be L. plantarum

5. Line 204: "2-chlorophenylalanine"?

6. Figure 1 could be improved. I think, there is no need to separate growth and pH curves. 

7. About Figure 2B, it is nothing new that L.plantarum strains can resist high salt concentrations.

8. The same opinion is in Table 1. This table should be represented as a supplementary file.

9. Line 387: "organoheterocyclic" ?

10. Line 406: should be Escherichia coli

11. Line 501: "isoleucyl and aspartate" ?

12. The discussion is well written, however, it could be improved to be more clearer and compact.

Reviewer 2 Report

Comments and Suggestions for Authors

The study isolated and screened an efficient strain of lactic acid bacteria from alfalfa silage, and analyzed the degradation ability and metabolic pathway of the strain on nitrite, which is innovative and has some guiding significance for alfalfa silage production and utilization. Although the overall logic of the whole paper is fair, there are some errors that need to be checked and corrected.

1.     Line 19 and 26: Is the degradation rate 99.06% or 98.48% within 24 h?

2.     Line 56-94: The section is overloaded with the findings of others and it is recommended that irrelevant content be deleted.

3.     Line 353:Figure 4 annotations?

4.     Line 403:Figure 7A and 7B?

5.     Line 482-524: To organize the discussion about nitrite degradation around the results and purpose of the study and to reduce the listing of other research results.

6.     Line 586-620: Are these metabolic pathways related to nitrite degradation in addition to the value-added metabolism of lactic acid bacteria?

7.     The effect of metabolic pathways on nitrite degradation was not mentioned in the conclusions.

8.     Line 631-632: Most of the metabolites are shown to show down-regulation in Figures 6 and 7, and how it can be seen that the levels of all these substances were found to be elevated?

9.     The first occurrence of an abbreviation requires an explanation of the full name.

10.  The language and logic of the article needs to be improved.

11. Please check the formatting of articles and references carefully to reduce unnecessary references.
